# The ubiquitin-like modifier FAT10 does not affect IL-12 expression and signaling

Jinjing Cao[1], Gerardo Omar Alvarez Salinas[1], Gunter Schmidtke[1], Michael Basler[1,2]*

1 Division of Immunology, Department of Biology, University of Konstanz, Konstanz, Baden-Württemberg, Germany, 2 Institute of Cell Biology and Immunology Thurgau (BITg), University of Konstanz, Kreuzlingen, Switzerland

* michael.basler@bitg.ch

## Abstract

The ubiquitin-like modifier FAT10 is strongly expressed in dendritic cells (DCs) and upregulated during inflammation. Interleukin (IL)-12 plays a critical role in promoting CD4$^+$ T cell differentiation into Th1 cells and in IFN-γ induction in T cells. Previously, it was shown that FAT10 is required for IFN-γ expression of activated T cells. In this study, we investigated whether FAT10 influences IL-12 expression or IL-12 induced signaling and thereby contributes to the reduced IFN-γ expression. Presence or absence of FAT10 did not alter IL-12 expression in DC2.4 cells and in bone marrow derived DCs. Furthermore, FAT10 had no influence on the differentiation of naïve T helper cells to Th1 cells under Th1 polarizing conditions. Additionally, FAT10 did not alter STAT4 phosphorylation in IL-12 receptor stimulated T cells. Taken together, FAT10 neither influences IL-12 expression in DCs nor affects IL-12 receptor signaling in T cells. Hence, the previously observed influence of FAT10 on IFN-γ secretion is not mediated by IL-12.

## Introduction

Whereas ubiquitin is expressed ubiquitously, human leukocyte antigen (HLA)-F adjacent transcript 10 (FAT10) is mainly expressed in tissues of the immune system such as lymph nodes or thymus [1,2]. Furthermore, FAT10 expression can be induced in a synergistic manner by IFN-γ and TNF [3], or during the maturation of dendritic cells (DCs) [1]. FAT10 is encoded in the major histocompatibility complex (MHC) locus [4,5], and is the only ubiquitin-like modifier that directly targets its substrates for degradation by the 26S proteasome in a ubiquitin-independent manner [6,7]. Hitherto, several proteins have been identified as interacting partners of FAT10, including the autophagy adaptor p62 [8], histone deacetylase 6 (HDAC6) [9], AP-1 transcription factor subunit (Jun) [10], proliferating cell nuclear antigen (PCNA) [11], β-catenin [12], mitofusin 2 (Mfn2) [13], FAT10 E3 ligase parkin [13], and the HECT-type ubiquitin E3 ligase HUWE1 [14]. Each of these proteins plays a critical role in different biological processes, underscoring the potential significance

**Data availability statement:** All relevant data are within the manuscript and its Supporting Information files.

**Funding:** This study was supported by the German Research Foundation (DFG) grant GR1517-27-1 to MB. Jinjing Cao is supported by China Scholarship Council (201906340171). The funders had no role in study design, data collection and analysis, decision to publish, or preparation of the manuscript.

**Competing interests:** The authors have declared that no competing interests exist.

of FAT10 in diverse cellular functions. Interestingly, phosphorylation of FAT10 seems to fine-tune FAT10's interactions with specific interaction partners, indicated by a more efficient conjugation to substrates of phosphorylated FAT10 [15]. Apart from different cellular functions, FAT10 is involved in different immunological processes [16,17]. Due to its selective expression in medullary thymic epithelial cells, it influences T cell selection [2]. Furthermore, FAT10 might play a role in the MHC class I antigen presentation pathway [16,18,19]. Moreover, FAT10 contributes to intracellular defense against bacteria by decorating autophagy-targeted *Salmonella* [20]. FAT10 knockout mice are viable and fertile, indicating that the lack of FAT10 does not interfere with essential housekeeping tasks [21]. However, lymphocytes of FAT10-deficient mice showed increased spontaneous apoptotic cell death. Infection of mice with influenza virus or lymphocytic choriomeningitis virus rapidly induces FAT10 mRNA expression [22]. Splenocytes from LCMV-infected FAT10-deficient mice exhibited diminished IFN-γ secretion and IL-12 p40 mRNA expression, while displaying enhanced production of type I interferons compared to their FAT10-proficient counterparts. Following viral infection, FAT10 fine-tunes the balance of interferons by reducing the production of type I interferons and increasing the levels of type II interferons [22].

Interleukin-12 (IL-12) is mainly secreted by dendritic cells (DCs), B cells, monocytes, and macrophages [23], and belongs to the IL-12 family, including other cytokines such as IL-23, IL-27, and IL-35 [24]. IL-12 consists of two subunits, IL-12p35 and IL-12p40, which are covalently linked to form the bioactive IL-12p70 [23]. The predominantly proinflammatory cytokine, plays a critical role in the development of T helper (Th)1 cells [24]. The signaling pathway of IL-12 involves the signal transducer activator of transcription (STAT) family, including STAT1, STAT4, and STAT5 [25]. Among these, STAT4 plays a key role in IL-12 receptor induced signaling.

In this study, we investigate whether altered IL-12 expression or signaling contribute to the previously observed reduced IFN-γ secretion of FAT10-deficient mouse CD8+ T cells.

## Materials and methods

### Mice

C57BL/6 mice were originally purchased from Charles River Laboratories (Sulzfeld, Germany). FAT10-/- mice [21] were kindly provided by A. Canaan and S. M. Weissman (Yale University School of Medicine, New Haven, CT). Mice were housed in a pathogen-free environment. They were matched by sex and age, and used when they were between 6–10 weeks of age. Mouse organ collection was approved by the Regierungspräsidium Freiburg ethics review board (approval number T-24/02TFA). Mice were euthanized using $CO_2$ inhalation, in accordance with approved animal care protocols. No experiments on living mice were performed.

### Cells

DC2.4 is a dendritic cell line (a kind gift from Dr. K. Rock, University of Massachusetts Medical School, Worcester, MA) and cultured in RPMI 1640 medium

(Gibco, 61870010), supplemented with 10% Fetal Bovine Serum (FBS, Gibco, 10270106), 1x Penicillin-Streptomycin (Gibco, 15140122). FAT10 expressing DC2.4 cells were generated by infecting DC2.4 cells with lentiviral particles encoding murine 3xFLAG-FAT10. After three days of cultivation, the cells were selected with 5 µg/ml puromycin (Gibco, A1113802).

To generate bone marrow-derived DCs (BMDCs), bone marrow from the femur and tibia of 10-weeks-old C57BL/6 wild-type mice and FAT10[-/-] mice was harvested. Isolated bone marrow cells were cultured in 10 cm dishes in RPMI 1640 medium, supplemented with 10% FBS, 1x penicillin/streptomycin, 50 µM β-mercaptoethanol (β-Me, ROTH, 4227.3), and 20 ng/ml recombinant murine GM-CSF (peprotech, 315–03) for 10 days.

### Lentivirus production

Production of lentiviruses was exactly performed as previously described [26]. In brief, lentiviral particles were produced by transient co-transfection of the expression vector 3xFlag-FAT10, the envelope vector pMD2.G, and the packaging vector psPAX2 into HEK293T cells using polyethylenimine (PEI, Polysciences, 23966). For this purpose, HEK293T cells were cultured in Iscove's Modified Dulbecco's Medium (IMDM) containing GlutaMAX (Gibco, 31980022) supplemented with 10% FCS, 100 U/ml penicillin, and 100 µg/ml streptomycin. Supernatant containing lentiviral particles was harvested 48 h and 72 h post-transfection. The lentiviral particles were stored at -80 °C.

### Th1 cell differentiation

CD4+ T cells were magnetically isolated from the spleens according to the manufacturer´s protocol (CD4 (L3T4) MicroBeads, Miltenyi Biotec, 130-117-043). Th1 differentiation was performed as previously described [27]. Shortly, purified cells were activated with plate-bound anti-mouse CD3ε antibody (BioLegend, 100302, Clone 145-2C11) and anti-mouse CD28 antibody (BioLegend, 102102, Clone 37.51) for 3 days in the presence of 10 ng/ml IL-12, 10 ng/ml IL-2, and 10 µg/ml anti-IL-4 (CytoBox Th1; Miltenyi Biotec, 130-107-761).

### Intracellular cytokine staining and flow cytometry

Polarized CD4+ T cells were restimulated for 5 h with 25 ng/ml phorbol 12-myristate 13-acetate (PMA) and 500 ng/ml ionomycin in the presence of 10 µg/ml brefeldin A. After surface staining with PE anti-mouse CD4 Antibody (BioLegend, 100408, Clone GK1.5), cells were fixed with 70 µl 4% paraformaldehyde, washed with permeabilization buffer (Invitrogen), and stained with FITC anti-mouse IFN-γ antibody (BioLegend, 505805, Clone XMG1.2). The samples were measured and analyzed on Accuri C6 flow cytometers (BD Biosciences, Franklin Lakes, New Jersey, USA).

### Immunoblotting

Cells were lysed in lysis buffer (20 mM Tris-HCl pH 7.6, 50 mM NaCl, 10 mM MgCl$_2$, 1% NP40, 1x EDTA-free Protease Inhibitor Cocktail (Roche, 4693132001), 1x PhosSTOP (Roche, 4906845001), incubated on ice for 30 min and centrifuged for 15 min at 13,000 x g. Total cell lysates were boiled with 4 × SDS sample buffer (250 mM Tris-HCl pH 6.8, 40% glycerol, 8% SDS, 0.004% bromophenol blue) supplemented with 20% β-mercaptoethanol at 95°C for 5 min. Equal amounts of cell lysates were separated by SDS-PAGE and transferred onto a nitrocellulose membrane. Protein was analyzed by the following antibodies: anti-pSTAT1 antibody (Cell Signaling, 7649S), anti-STAT1 antibody (Cell Signaling, 9172S), and anti-pSTAT4 antibody (Cell Signaling, 4134S). Anti-GAPDH antibody (Sigma-Aldrich, G9545) served as loading control. IRDye680RD goat anti-mouse (LICOR, 926-68070) or anti-rabbit (LICOR, 926-68071) and IRDye800CW goat anti-rabbit (LICOR, 926-32211) or anti-mouse (LICOR, 926-32210) were used as secondary antibodies. The LI-COR Odyssey Imager (LI-COR Biosciences, Lincoln, Nebraska, USA) and the Image Studio Lite Version 5.2 were used for analyzing signals.

### Real-time RT-PCR

Total RNA was extracted (RNeasy Mini Kit, QIAGEN, 74106) from cells, followed by reverse transcription (cDNA Synthesis Kit, Biozym, 331470L). Then, real-time RT-PCR (Biozym Blue S´Green Kit, 331416XL) was executed in a Biometra TProfessional Thermocycler (Analytik Jena). The quantitative value of each sample was normalized to mouse GAPDH, which was used as reference gene. Further information on primer pairs and reagents can be found in Supplementary Tables S1 and S2.

### ELISA

The secretion of IL-12 was measured according to the protocol provided by the manufacturer (ELISA MAX™ Deluxe Set Mouse IL-12 (p70), BioLegend, 433604).

### Statistical analysis

Statistical analysis was conducted using Prism software (GraphPad, version 8.0). For comparisons involving two continuous variables, a two-tailed unpaired Student's t-test was used, while for comparisons involving more than two continuous variables, one-way ANOVA followed by Tukey's multiple comparison test was applied. The data are presented as individual scattered points along with the mean ± standard deviation (SD).

## Results

### FAT10 does not influence IL-12 expression in DC2.4 cells

Mah et al. showed that in the absence of FAT10 IFN-γ secretion of TCR stimulated splenocytes derived from LCMV-infected mice is strongly reduced [22]. Furthermore, these splenocytes expressed less mRNA for IL-12 p40 but secreted more IFN-α and IFN-β. Il-12 secreted by DCs is driving the differentiation of naïve T helper cells into Th1 cells and contributes to the proper activation of cytotoxic T cells. In this study, we intend to investigate the effect of FAT10 on IL-12 expression and signaling. In a first step, the effect of FAT10 on IL-12 expression was investigated in the murine dendritic cell-like cell line DC2.4. Since there are no antibodies available recognizing mouse FAT10 protein, FAT10 expression levels had to be analyzed on mRNA level using real-time RT-PCR. Non-stimulated DC2.4 cells barley express FAT10 mRNA (Fig 1A). Stimulation of DC2.4 with IFN-γ or IFN-γ/TNF strongly induced FAT10 mRNA, whereas LPS alone had no effect on FAT10 expression. Analysis of IL-12 mRNA levels in these stimulated cells, revealed a strong IL-12 mRNA induction (Fig 1B). Hence, DC2.4 is a suitable cell line to investigate the effect of FAT10 on IL-12. DC2.4 cells were stably transfected with FAT10 (Fig 1C). Stimulation with LPS strongly induced IL-12 mRNA in DC2.4 wild type and FAT10 overexpressing cells. However, there was no difference in IL-12 mRNA level in FAT10 overexpressing DC2.4 cells compared to wild type DC2.4 cells.

### FAT10 has no significant effect on IL-12 expression in bone marrow-derived dendritic cells

To further explore the effect of FAT10 on IL-12, we conducted experiments using bone marrow-derived dendritic cells (BMDCs) from C57BL/6 wild type mice and FAT10$^{-/-}$ mice. To up-regulate FAT10, in vitro generated BMDCs were stimulated with IFN-γ for one day to induce FAT10 expression. TNF was not included in this stimulation setup to prevent IL-12 expression during FAT10 induction. This was followed by an additional day of LPS stimulation to induce IL-12 expression. Indeed, in this experimental set-up we observed a strong induction of FAT10 (Fig 2A) and IL-12 (Fig 2B). However, FAT10 wild type and FAT10-deficient BMDCs induced similar levels of IL-12 mRNA. Next, we analyzed IL-12 secretion into the supernatant of IFN-γ/LPS stimulated wild type and FAT10-deficient BMDCs by ELISA. Similar to mRNA levels, FAT10 did not affect IL-12 secretion in BMDCs (Fig 2C).

### FAT10 does not affect Th1 cell differentiation

IL-12 is essential for Th1 cell differentiation, which plays a crucial role in protective cell-mediated immunity against various intracellular pathogens [28,29]. In vitro, Th1 cells can be generated in the presence of IL-12. If FAT10 plays a crucial role

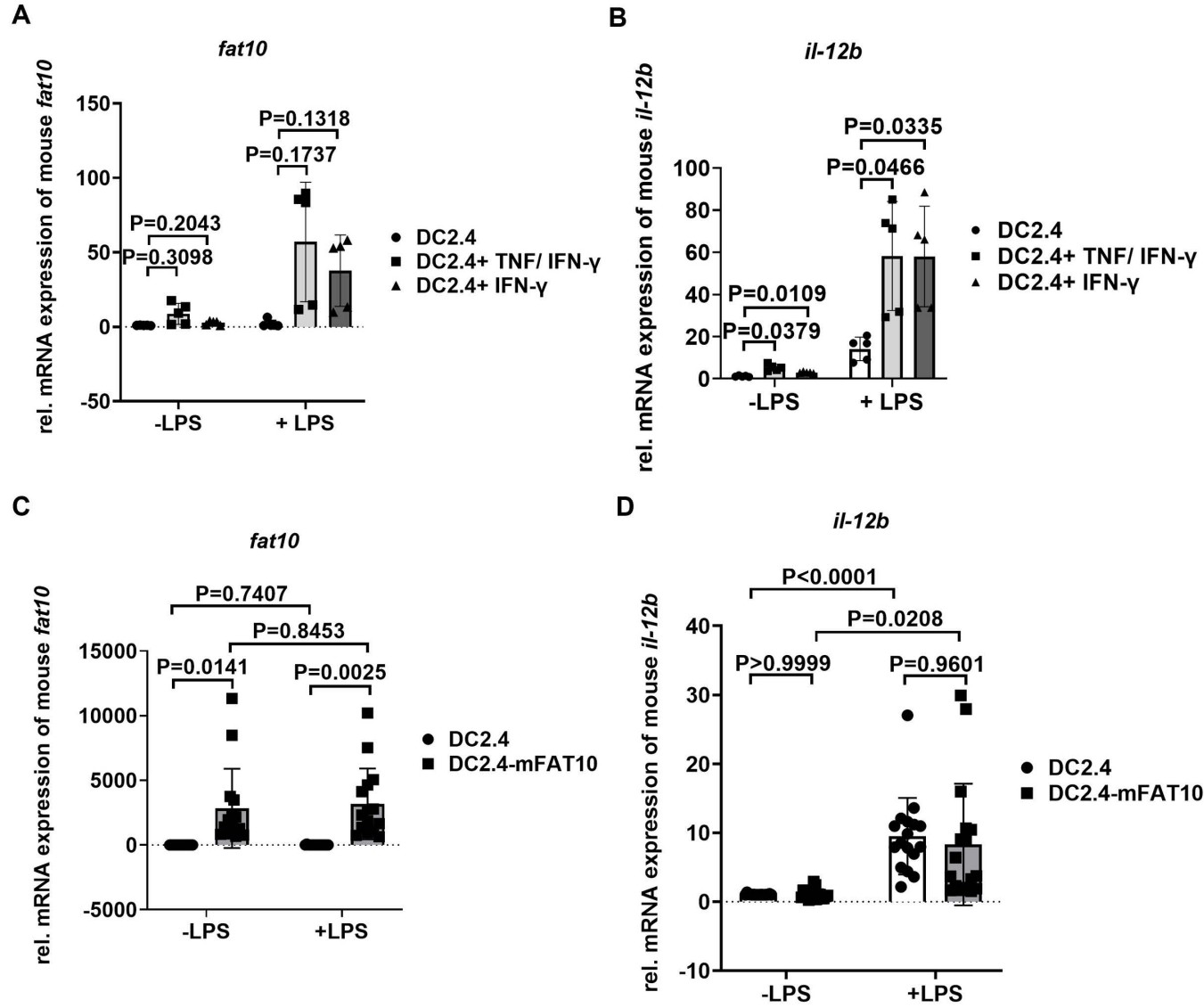

**Fig 1. The Impact of FAT10 on IL-12 expression in DC2.4 cells.** (A, B) DC2.4 cells were treated with TNF (400 U/ml)/IFN-γ (200 U/ml), IFN-γ (200 U/ml) in combination with (indicated +LPS) or without (indicated -) LPS (2 µg/ml) for 1 day. The mRNA level of FAT10 (A) and IL-12 (B) were detected by real-time RT-PCR. (C, D) DC2.4 cells and DC2.4-mFAT10 cells (stably expressing mouse FAT10) were treated with (indicated +LPS) or without (indicated -) LPS (2 µg/ml) for 1 day. The mRNA level of FAT10 (C) and IL-12 (D) were detected by real-time RT-PCR. (A-D) The y-axis depicts relative mRNA levels normalized to GAPDH. Data was analyzed by one-way ANOVA with Tukey's multiple comparisons test. P values are indicated. Data are depicted as mean±SDs. Each symbol in the graph represents an individual experiment. A, B: n=5; C: n=17, D: n=15.

in response to IL-12 stimulation, this should affect differentiation of FAT10-deficient Th1 cells. CD4+ T cells were magnetically isolated from the spleens of C57BL/6 wild type mice and FAT10-/- mice and cultured under Th1-polarizing conditions for 72 hours. IFN-γ is the signature cytokine of Th1 cells. Therefore, efficiency of Th1 differentiation was analyzed by intracellular IFN-γ staining of CD4+ T cells. No significant differences in Th1 differentiation between C57BL/6 wild type and FAT10-/- CD4+ T cells could be observed (Fig 3), indicating that FAT10 does not play an important role in the Th1 skewing process.

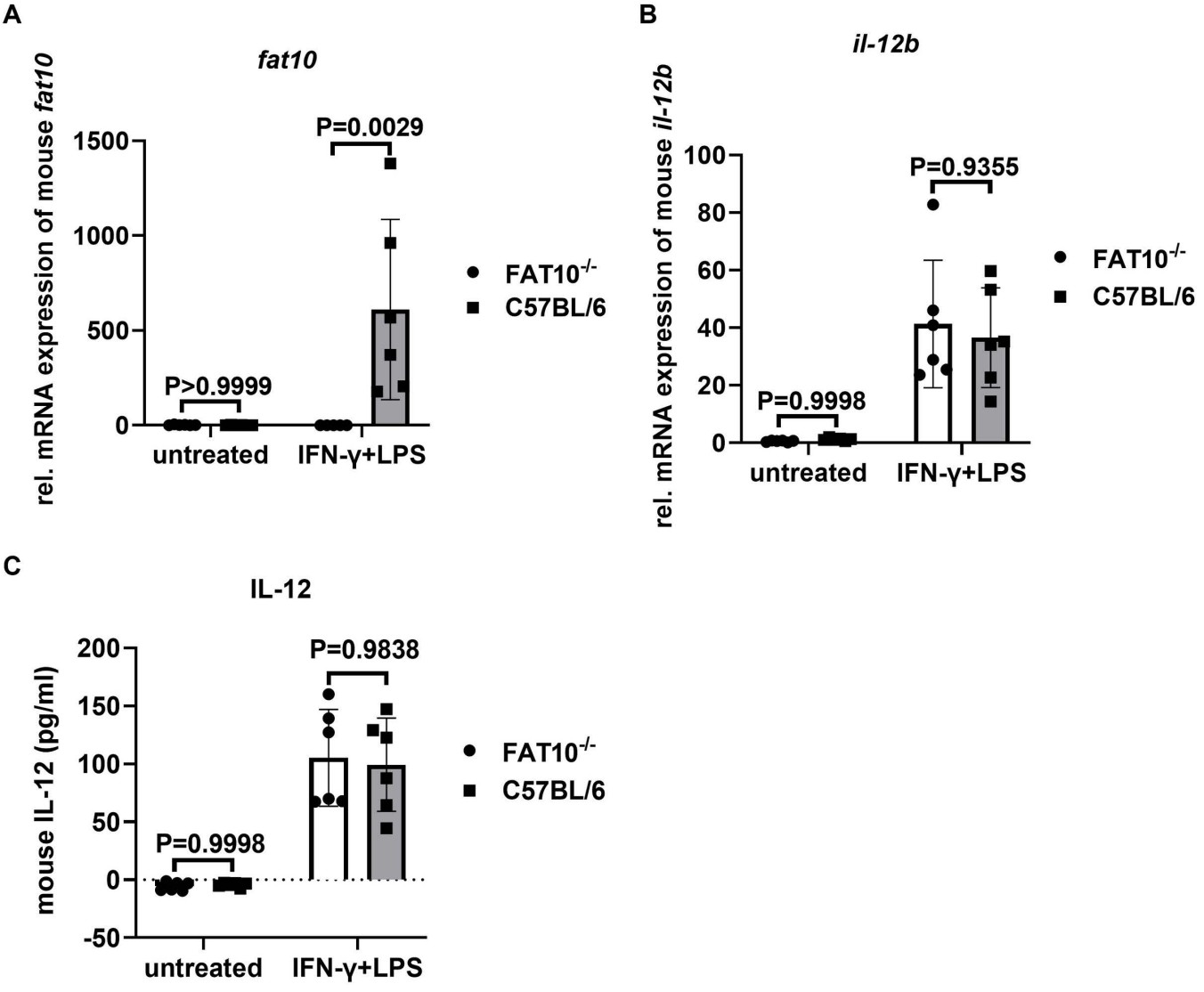

**Fig 2. No impact of FAT10 on IL-12 expression in BMDCs.** (A-C) BMDCs were generated from C57BL/6 wild type mice or FAT10⁻/⁻ mice. BMDCs were treated with IFN-γ (200 U/ml) for 1 day, followed by LPS (1 μg/ml) treatment for 1 further day (indicated IFN-γ+LPS) or were left untreated (indicated untreated). (A, B) The mRNA level of FAT10 (A) and IL-12 (B) were detected by real-time RT-PCR. The y-axis depicts relative mRNA levels normalized to GAPDH. (C) The secretion of IL-12 into the supernatant was measured by ELISA. (A-C) Data was analyzed by one-way ANOVA with Tukey's multiple comparisons test. P values are indicated. Data are depicted as mean±SDs. Each symbol in the graph represents BMDCs derived from an individual mouse. n=6 per group.

## FAT10 does not modulate STAT signaling pathway

Although the absence of FAT10 had no crucial role on Th1-polarizing condition, we further investigated the effect of FAT10 in IL-12 receptor signaling. IL-12 activates CD4⁺ T cells to produce IFN-γ via the STAT signaling pathway [30]. After IL-12 interacts with its receptor subunits IL-12Rβ1 and IL-12Rβ2, it triggers two signaling pathways: the Janus kinase (JAK) and STAT pathways [31,32]. STAT4 serves as the primary downstream signaling target of IL-12, whereas IL-12 has less influence on STAT1, STAT3, and STAT5 molecules [28]. Therefore, we only investigated STAT4 and STAT1. Murine naïve T cells barely express FAT10 [33]. Therefore, magnetically isolated CD4⁺ T were stimulated with TNF/IFN-γ for 1 day.

**A**

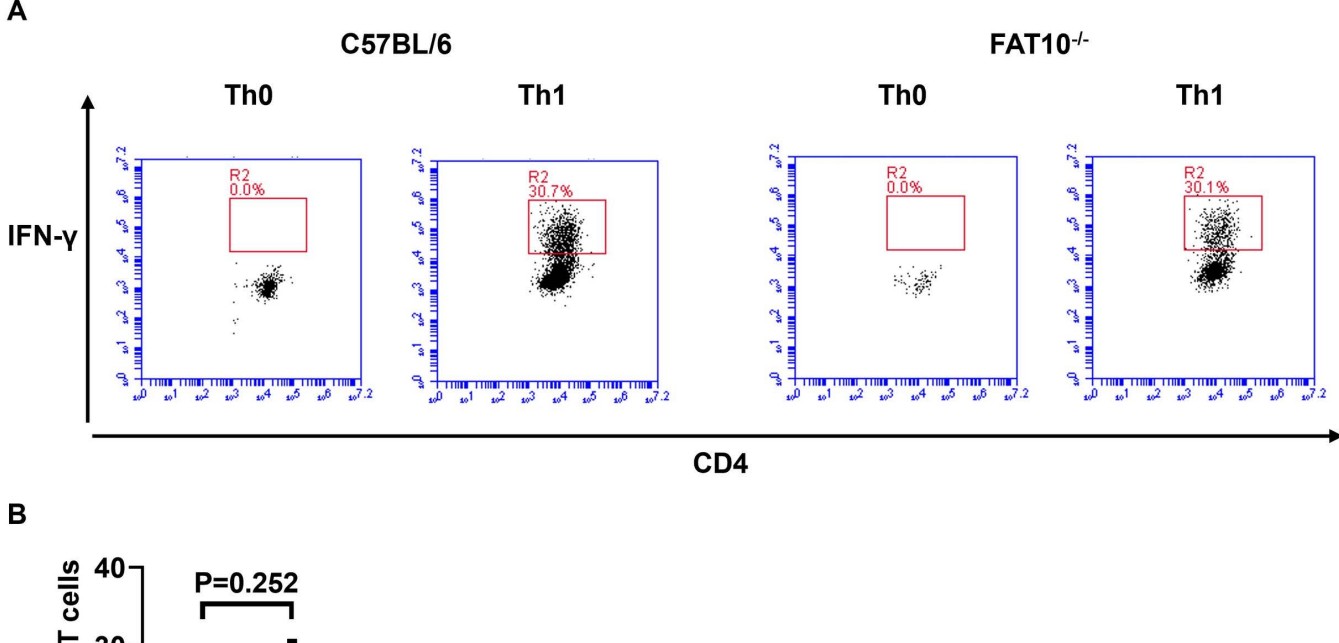

**B**

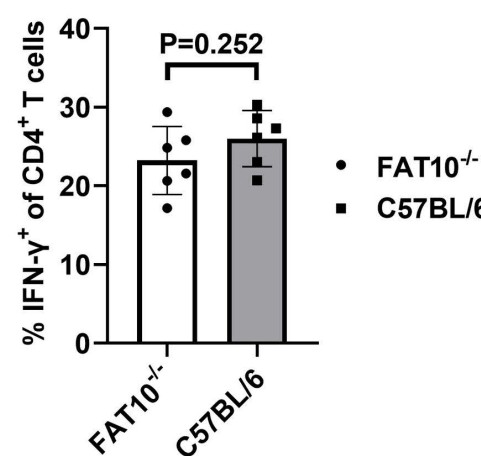

**Fig 3. FAT10 does not affect Th1 differentiation.** CD4+ T cells were magnetically purified from the spleen of C57BL/6 wild type mice or FAT10−/− mice. Purified cells were cultured under Th0 or Th1 skewing condition for 72h. Cells were restimulated with PMA/ionomycin in the presence of brefeldin A for 4h, stained for CD4 and intracellular IFN-γ, and analyzed by flow cytometry. (A) Representative dot plots. (B) Bar graph depicting percentage of IFN-γ+CD4 cells. Data are depicted as mean±SDs of CD4+ cells isolated from 6 different mice per group (n=6) statistically analyzed by unpaired Student's *t*-test.

Analysis of STAT1 phosphorylation by western blot revealed that this treatment induced the pSTAT1 (Fig 4A). Hence, this experimental set-up could not be used to investigate activation of STAT1 after IL-12R stimulation. Nevertheless, we observed no influence of FAT10 on STAT1 phosphorylation after TNF/IFN-γ stimulation (Fig 4A,B). After IL-12R stimulation, STAT4 is phosphorylated, and dimerization enables subsequent nuclear translocation. Magnetically isolated CD4+ T cells were treated with TNF/IFN-γ to induce FAT10 for 24h [33]. Interestingly, this treatment did not induce STAT4 phosphorylation (Fig 4C). TNF/IFN-γ pre-treated cells were stimulated with IL-12 for 2h. Phosphorylation of STAT4 was analyzed by western blot. Within 2h, pSTAT4 could be detected. However, no difference in STAT4 phosphorylation could be observed between C57BL/6 wild type and FAT10-deficient CD4+ T cells (Fig 4D). These results suggest that FAT10 does not modulate the STAT signaling pathway in CD4+ T cells.

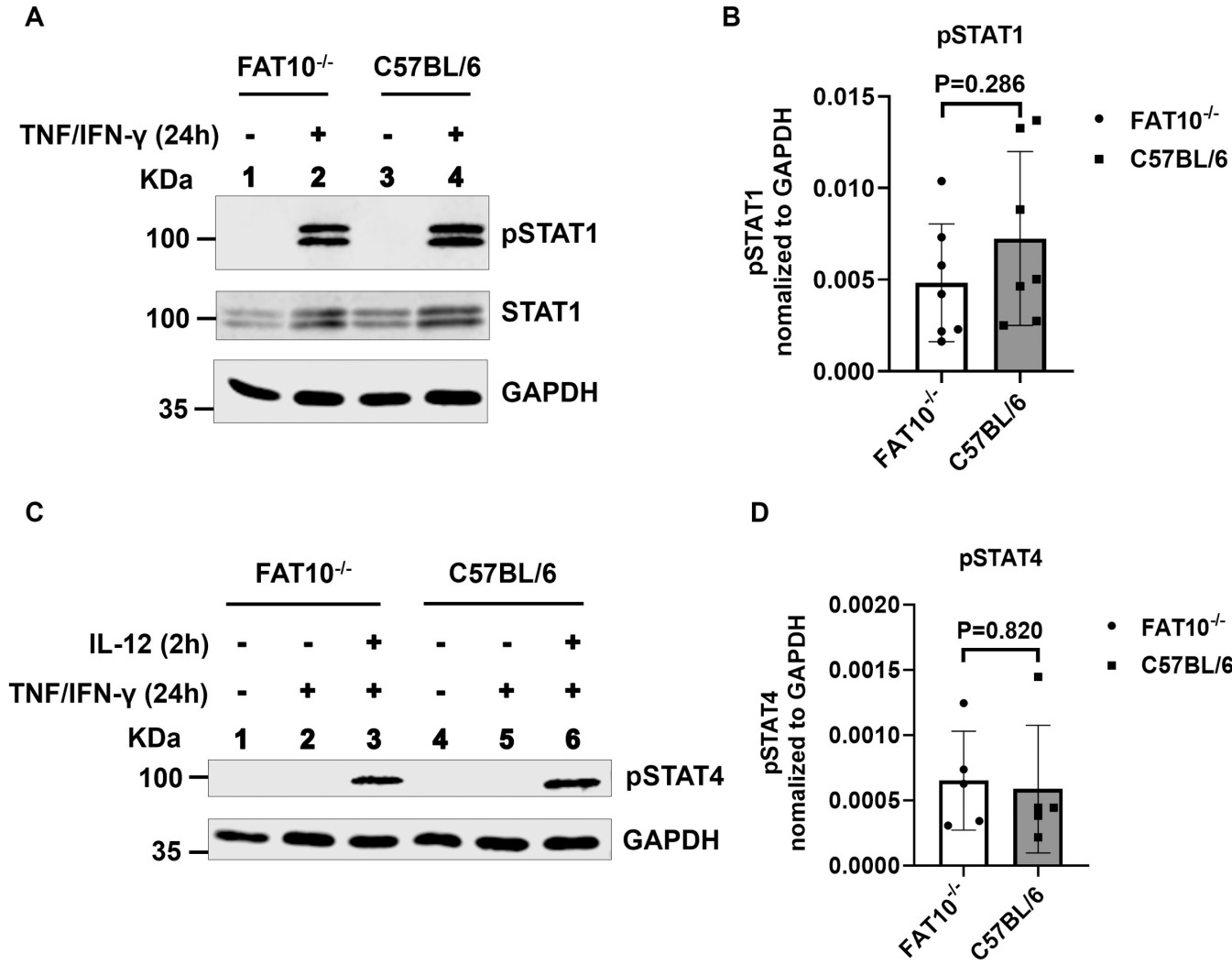

**Fig 4. FAT10 does not modulate the IL-12 signaling pathway.** (A-D) CD4[+] T cells were magnetically purified from spleens of C57BL/6 wild type mice or FAT10[-/-] mice. (A, B) CD4[+] T cells were treated with TNF (400 U/ml)/IFN-γ (200 U/ml) (indicated +) for 1 day or were left untreated (indicated -). pSTAT1, STAT1, and GAPDH were analyzed by western blot. (B) Quantification of pSTAT1 normalized to GAPDH. Data are depicted as mean±SDs derived from 7 different experiments (n=7). (C, D) CD4[+] T cells were treated with TNF (400U/ml)/IFN-γ (200 U/ml) for 1 day or left untreated, followed by a treatment with IL-12 (10 ng/ml) for 2 h. Treatment regime is indicated. pSTAT4 and GAPDH were analyzed by western blot. (D) Quantification of pSTAT4 normalized to GAPDH. Data are depicted as mean±SDs derived from 5 different experiments (n=5).

## Discussion

FAT10 is the only ubiquitin-like modifier targeting proteins for degradation by the proteasome [7]. By this mean it can potentially control any processes, such as signaling or transcription, within cells. Since FAT10 is expressed in lymphoid tissues and can be strongly induced in an inflammatory environment, such processes might play a crucial role in inflammation. Indeed, phosphorylated FAT10 is essential for the efficient inhibition of IFN-β secretion upon TNF stimulation and during influenza A virus infection [34]. Phosphorylated FAT10 has been demonstrated to interact with OTUB1, leading to a reduction in whole ubiquitylation levels and suppression of IFN-I induction through modulation of TRAF3 signaling. Furthermore, during influenza A virus infection, the presence of FAT10 leads to a reduction in the total ubiquitination of the E3

ligase TRIM21. The degradation of TRIM21 mediated by FAT10 decreased IFN-β production.[35]. Dendritic cells initiate antigen-specific immune responses and are considered as the master regulators of the immune response [36]. FAT10 is strongly expressed in human monocyte-derived dendritic cells [1]. Notably, FAT10 mRNA expression is increased during the later stages of dendritic cell maturation. Dendritic cells stimulated with LPS exhibited enhanced FAT10 expression and an accumulation of FAT10 conjugates [37]. IL-12 is primarily secreted by monocytes, macrophages, B cells and dendritic cells [23]. IL-12 secreted by DCs is crucial for proper activation of cytotoxic T cells and T helper cells and for promoting Th1 differentiation [38]. Interestingly, FAT10$^{-/-}$ mice exhibit higher IL-12p40 expression in the spleen compared to FAT10$^{+/-}$ mice three days post LCMV infection [22]. This raised the question whether FAT10 is involved in regulating IL-12 expression in dendritic cells or influencing IL-12 receptor signaling in T cells.

The murine DC cell line DC2.4 is an established model to study DC functions [39]. DC maturation stimuli, such as LPS, poly (I:C), IL-1β, and R848 can induce IL-12 production [39–41]. In a first step, we investigated whether LPS can induce IL-12 mRNA in DC2.4. Indeed, LPS induced IL-12p40 mRNA 24 h post stimulation (Fig 1B). IL-12p35 mRNA induction could not consistently and reliably be detected in our set-up, which might slightly limit the interpretation of our results. Whereas LPS barely induced FAT10, IFN-γ or TNF/IFN-γ strongly upregulated FAT10 expression in DC2.4 cells (Fig 1A). Interestingly, IFN-γ or TNF/IFN-γ treatment strongly enhanced LPS induced IL-12p40 mRNA expression. To study the effect of FAT10 on IL-12p40 mRNA induction in DC2.4 cells, DC2.4 cells stably overexpressing FAT10 were generated (Fig 1C). Notably, wild type DC2.4 cells barely express FAT10. LPS stimulation induced a strong induction of IL-12p40 mRNA in wild type and FAT10 overexpressing DC2.4 cells. However, absence of FAT10 (wild type DC2.4 cells) or overexpression of FAT10 had no influence on IL-12p40 mRNA expression. These results were confirmed in primary bone marrow derived dendritic cells derived from wild type C57BL/6 mice and FAT10-deficient mice (Fig 2). Although FAT10 was strongly up-regulated in wild type LPS/IFN-γ stimulated BMDCs (Fig 2A), no influence of FAT10 on IL-12 mRNA (Fig 2B) or protein (Fig 2C) levels could be observed in BMDCs. Taken together, these data strongly indicate that FAT10 does not play a significant role in IL-12 expression in murine DCs (Figs 1,2).

Mah et al. reported a reduced IFN-γ production of TCR-stimulated FAT10-deficient splenocytes [22]. The authors speculated that altered IL-12 expression could account for the diminished IFN-γ production in the absence of FAT10. However, we demonstrated that IL-12 expression is not altered in presence or absence of FAT10 (Figs 1,2). IL-12 stimulation of T cells strongly contributes to IFN-γ expression of activated T cells [42]. Hence, an altered IL-12 signaling in T cells might also lead to the reduced IFN-γ expression observed by Mah et al. [22]. Since IL-12 stimulates CD4$^+$ T cell differentiation into Th1 cells via the STAT signaling pathway, the influence of FAT10 on Th1 differentiation was investigated. Wild type and FAT10-deficient CD4$^+$ T cells were polarized in Th1 skewing condition. Both wild type and FAT10$^{-/-}$ CD4$^+$ cells differentiated to IFN-γ producing Th1 cells. However, no influence of FAT10 on Th1 differentiation was observed (Fig 3). This indicates that the absence of FAT10 in T cells does influence IL-12R signaling induced by the Th1 skewing cytokine IL-12. Although the requirement of IL-12 for Th1 differentiation is well established, we can not completely exclude that IL-12 is required at all for Th1 differentiation in our in vitro Th1 differentiation set-up, which might limit the interpretation of our results. Nevertheless, IL-12 receptor signaling was further investigated in T cells (Fig 4). However, IL-12 induced signaling in CD4$^+$ T cells manifested in similar phosphorylation of STAT4 in FAT10-proficient and FAT10-deficient cells, showing that FAT10 does not influence IL-12R signaling in CD4$^+$ T cells.

Taken together, we found no indications that FAT10 alters IL-12 expression in DCs nor IL-12 induced signaling in T cells. Therefore, we conclude that a different mechanism is responsible for the previously observed reduction of IFN-γ secretion in FAT10-deficient splenocytes [22].

## Supporting information

**S1 Table. Primer pairs for used for real-time RT PCR.**
(TIF)

**S2 Table. Reagents.**
(TIF)

**S1 Fig. Flow cytometry gating strategy of** Fig 3.
(TIF)

**S2 Fig. The *fat10* gene expression in C57BL/6 mice of** Fig 4**. CD4+ T cells were magnetically purified from spleens of C57BL/6 wild type mice.** CD4+ T cells were treated with TNF (400 U/ml)/IFN-γ (200 U/ml) (indicated +) for 1 day or were left untreated (indicated -). The mRNA level of FAT10 were determined by real-time RT-PCR. The y-axis depicts relative mRNA levels normalized to GAPDH. Data are depicted as mean ± SDs derived from 4 different experiments (n = 4).
(TIF)

**S3 Fig. Original uncropped western blots of** Fig 4.
(TIF)

## Acknowledgments

We thank Katharina Inholz, Natalie Pach, Dennis Mink, Dennis Horvath, Christine Wünsch, and Ilona Kindinger for their continued help.

## Author contributions

**Conceptualization:** Jinjing Cao, Michael Basler.

**Data curation:** Jinjing Cao, Michael Basler.

**Formal analysis:** Jinjing Cao, Michael Basler.

**Methodology:** Jinjing Cao, Gerardo Omar Alvarez Salinas, Gunter Schmidtke, Michael Basler.

**Supervision:** Gunter Schmidtke, Michael Basler.

**Writing – original draft:** Jinjing Cao.

**Writing – review & editing:** Michael Basler.

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
