## [Decision Letter · Decision Letter 0]

24 Feb 2025

PONE-D-25-01617The ubiquitin-like modifier FAT10 does not affect IL-12 expression and signalingPLOS ONE

Dear Dr. Basler,

Thank you for submitting your manuscript to PLOS ONE. After careful consideration, we feel that it has merit but does not fully meet PLOS ONE’s publication criteria as it currently stands. Therefore, we invite you to submit a revised version of the manuscript that addresses the points raised during the review process.

We look forward to receiving your revised manuscript.

Kind regards,

Mariola J Ferraro, Ph.D.

Academic Editor

PLOS ONE

2. To comply with PLOS ONE submissions requirements, in your Methods section, please provide additional information regarding the experiments involving animals and ensure you have included details on (1) methods of sacrifice, and (2) efforts to alleviate suffering.

https://c.coek.info/pdf-maturation-of-human-dendritic-cells-is-accompanied-by-functional-remodelling-of-.html

https://www.life-science-alliance.org/content/7/1/e202101282?rss=1

https://www.excli.de/vol19/2020-3104/2020-3104.htm

https://www.researchgate.net/publication/369773712_The_ubiquitin-like_modifier_FAT10_is_degraded_by_the_20S_proteasome_in_vitro_but_not_in_cellulo

In your revision ensure you cite all your sources (including your own works), and quote or rephrase any duplicated text outside the methods section. Further consideration is dependent on these concerns being addressed.

“German Research Foundation (DFG) grant GR1517-27-1 to MB.”

Reviewers' comments:

Reviewer's Responses to Questions

**Comments to the Author**

1. Is the manuscript technically sound, and do the data support the conclusions?

Reviewer #1: Partly

Reviewer #2: Partly

2. Has the statistical analysis been performed appropriately and rigorously? 

Reviewer #1: No

Reviewer #2: Yes

3. Have the authors made all data underlying the findings in their manuscript fully available?

Reviewer #1: Yes

Reviewer #2: Yes

4. Is the manuscript presented in an intelligible fashion and written in standard English?

Reviewer #1: Yes

Reviewer #2: Yes

5. Review Comments to the Author

Reviewer #1: The study investigates the impact of FAT10 on IL-12 expression and signaling, which has not been extensively studied before. They do this in two very specific models of immune cells. There is no indication of prior publication of these specific results. But this may be a follow up to previous results. The experiments are well-described, but some analyses (e.g., small sample sizes in Fig. 4, lack of flow cytometry gating strategy in figure 3) could be improved for better rigor. Increasing sample size or better more obvious normalization will improve figures like figure one where you see a separation of points which may be a reasoning for expression to not be different. Evaluating fat10 gene expression in C57/B6 mice in figure 4 would help the reader believe that fat10 expression is still up during this window of time. Did the authors do a time course experiment to show why they chose 24 hours of stimulation? It is also not scientifically explained why the authors would shift away from the TNF/IFNg model of expression seen in figure 1A. Yes, conclusions are presented in an appropriate fashion but with limitations. The data show makes assumptions about expression which would support the conclusion that FAT10 does not affect IL-12 expression or signaling. However, variability in some experiments and small sample sizes in signaling studies limit the strength of the conclusions. Including more controls and sometimes replicates would improve the results presented in this study to ensure that the study is properly powered and controlled.

Major Comments

1. Clarity in Experimental Design

In Th1 differentiation assays, I do not think they included control. it would be beneficial to include a control condition without IL-12 to demonstrate the necessity of IL-12 in this system.

2. STAT Signaling Pathway Investigation (line 208)

The authors analyze STAT1 and STAT4 phosphorylation, but IL-12 is also known to activate STAT3 to a certain extent. They briefly stated this without clearly justifying why they did not look at STAT3

3. IL-12 Expression and Secretion (line 244)

Figure 2C shows that IL-12 secretion is not altered in FAT10 knockout cells. Since IL-12 has two subunits (p35 and p40), the authors did not measure p35 expression separately, and did not create a limitation section to acknowledged this as a limitation.

4. Potential Redundancy in Text

The authors repeatedly emphasize that FAT10 does not alter IL-12 expression across different sections (Results, Discussion).

Conclusion

The manuscript is well-conceived and presents valuable findings in immunology and cytokine signaling. With minor revisions to improve clarity, correct typographical errors, and address the misplaced reference, the manuscript will be well-suited for publication.

Key Recommendations

Address the STAT3 signaling question.

Consider IL-12 p35 measurement.

Remove redundant statements in Discussion.

not should be no in line 214.

Reviewer #2: While the introduction is technically sound, I feel a re-write may be in order to emphasize the importance of this research and how it's applications may benefit immunological research. That issue aside, there were several factors that warranted my recommendation for major revision in Results section of this manuscript. On line 169 under the "FAT10 does not influence IL-12 expression in DC2.4 cells" segment, there didn't appear to be any mention of validating successful lentiviral transfection. In the next segment "FAT10 has no significant effect on IL-12 expression in bone marrow-derived dendritic cells", BMDCs were stimulated with IFN-γ and LPS to induce IL-12 expression. However, in the previous segment a combination IFNγ/TNF was used to stimulate DC2.4s. The rationale behind using only IFN-γ as opposed to the combination IFNγ/TNF wasn't immediately clear upon review. Lastly, in "FAT10 does not affect Th1 cell differentiation", the rationale as to why IFN-γ was selected as the sole cytokine of focus was unclear. TNF is also considered an important signature cytokine for Th1 differentiation, why was that not selected in addition to IFN-γ to validate results? Overall, while I do partly agree with the overall conclusions of this paper that FAT10 does not appear to affect IL-12 signaling, I feel more validatory work is necessary to confirm many of these results.

6. PLOS authors have the option to publish the peer review history of their article (what does this mean? ). If published, this will include your full peer review and any attached files.

**Do you want your identity to be public for this peer review?** For information about this choice, including consent withdrawal, please see our Privacy Policy .

Reviewer #1: No

Reviewer #2: No

---

## [Author Response · Author response to Decision Letter 0]

21 Mar 2025

Response to the reviewer comments you will find in the uploaded "Response to reviewers" file.

5. Review Comments to the Author

Please use the space provided to explain your answers to the questions above. You may

also include additional comments for the author, including concerns about dual

publication, research ethics, or publication ethics. (Please upload your review as an

attachment if it exceeds 20,000 characters)

Reviewer #1: The study investigates the impact of FAT10 on IL-12 expression and

signaling, which has not been extensively studied before. They do this in two very

specific models of immune cells. There is no indication of prior publication of these

specific results. But this may be a follow up to previous results. The experiments are welldescribed, but some analyses (e.g., small sample sizes in Fig. 4, lack of flow cytometry

gating strategy in figure 3) could be improved for better rigor.

Reply: We agree that sample size in Fig. 4 is rather low. Therefore, we increased

sample size in Fig. 4. Furthermore, we included the gating strategy of Figure 3 (in

supplementary Fig. S1).

Increasing sample size or better more obvious normalization will improve figures like

figure one where you see a separation of points which may be a reasoning for expression

to not be different.

Reply: Real-time RT-PCR experiments were performed in Figure 1. Each data point

represents the average of duplicates. The cycle threshold (Ct) value was obtained

from the experiment. Gene expression was calculated using the 2−ΔΔCt method,

where ΔCt = Ct (target gene) - Ct (reference gene), and ΔΔCt = ΔCt (treated sample)

- ΔCt (control sample). All data were already normalized to untreated DC2.4 cells.

Evaluating fat10 gene expression in C57BL/6 mice in figure 4 would help the reader

believe that fat10 expression is still up during this window of time.

Reply: We included the FAT10 expression data of CD4+ cells from C57BL/6 mice

stimulated for 1 day with IFNγ/TNF in the revised manuscript (supplementary Fig.

S2). Furthermore, we have repeated the Western blot experiments for Figure 4 to

increase sample size.

Did the authors do a time course experiment to show why they chose 24 hours of

stimulation?

Reply: Initially, when setting up the experiments, we tried different time points and

compared gene expression levels with STAT activation. If the incubation time was

too short, FAT10 gene expression was difficult to detect. If the incubation time was

too long, the culture medium turned yellow, indicating that culture conditions were

suboptimal. Finally, we chose 24 hours, a time point FAT10 expression was readily

detected and STAT activation with IL-12 could be observed.

It is also not scientifically explained why the authors would shift away from the TNF/IFNg

model of expression seen in figure 1A.

Reply: In Figure 3, we only used IFNγ stimulation without TNF stimulation for the

following reason: To investigate whether FAT10 plays a role in IL-12 secretion in

BMDCs, we need FAT10 expression. Therefore, cells were stimulated with IFNγ for

24 h. After the induction of FAT10, IL-12 is induced with LPS. If we do the initial

FAT10 induction in the presence of TNF, TNF will mature BMDCs and therefore will

induce the expression of IL-12. However, to investigate a contribution of FAT10 inIL-12 induction, we first need a FAT10 expression. As seen in Fig. 2A, FAT10

expression in BMDCs is also strongly induced in the absence of TNF. We

explained this in the revised manuscript

Yes, conclusions are presented in an appropriate fashion but with limitations. The data

show makes assumptions about expression which would support the conclusion that

FAT10 does not affect IL-12 expression or signaling. However, variability in some

experiments and small sample sizes in signaling studies limit the strength of the

conclusions. Including more controls and sometimes replicates would improve the results

presented in this study to ensure that the study is properly powered and controlled.

Reply: We increased the sample size in Figure 4 to ensure the experiment is

properly powered. In our opinion, the other experiments with n≥5 are properly

powered.

Major Comments

1. Clarity in Experimental Design

In Th1 differentiation assays, I do not think they included control. it would be beneficial to

include a control condition without IL-12 to demonstrate the necessity of IL-12 in this

system.

Reply: It is well established that IL-12 is required for Th1 differentiation (f.i.

Trinchieri G. Annu Rev Immunol. 1995;13:251-76, Trinchieri G. Blood.

1994;84(12):4008-27, Neurath et al., J Exp Med. 1999;182(5):1281-90, Nakamura et

al., J Immunol. 1997;158(3):1085-94, etc.). Therefore, we did not include this

control. Nevertheless, we discuss this as a limitation of our study and mainly

conclude that FAT10 does not influence Th1 differentiation.

2. STAT Signaling Pathway Investigation (line 208)

The authors analyze STAT1 and STAT4 phosphorylation, but IL-12 is also known to

activate STAT3 to a certain extent. They briefly stated this without clearly justifying why

they did not look at STAT3.

Reply: See below.

3. IL-12 Expression and Secretion (line 244)

Figure 2C shows that IL-12 secretion is not altered in FAT10 knockout cells. Since IL-12

has two subunits (p35 and p40), the authors did not measure p35 expression separately,

and did not create a limitation section to acknowledged this as a limitation.

Reply: See below.

4. Potential Redundancy in Text

The authors repeatedly emphasize that FAT10 does not alter IL-12 expression across

different sections (Results, Discussion).

Reply: See below.

Conclusion

The manuscript is well-conceived and presents valuable findings in immunology and

cytokine signaling. With minor revisions to improve clarity, correct typographical errors,

and address the misplaced reference, the manuscript will be well-suited for publication.

Reply: We have updated the references in our manuscript.Key Recommendations

1. Address the STAT3 signaling question.

Reply: The study focuses on the relationship between FAT10 and IL-12 in Th1

cells. The signaling pathway related to Th1 cells are mainly the STAT4 and STAT1

signaling pathways (Trinchiere, Nat Rev Immunol. 2003 Feb;3(2):133-46). STAT3

plays a minor role. Therefore, we only investigated STAT1 and STAT4 signaling.

We indicate this in the revised manuscript.

2. Consider IL-12 p35 measurement.

Reply: We performed real-time RT-PCR to detect the expression levels of IL-12p40

and IL-12p35 in DC2.4 cells. However, the cycle threshold (Ct) value for IL-12p35

could not be reliably measured in our hands. Therefore, we only measured IL-

12p40. We discuss this as a limitation in the revised manuscript.

3. Remove redundant statements in Discussion.

Reply: We tried to remove redundant statements in the discussion of the revised

manuscript.

4. not should be no in line 214.

Reply: We changed this.

Reviewer #2: While the introduction is technically sound, I feel a re-write may be in order

to emphasize the importance of this research and how it's applications may benefit

immunological research. That issue aside, there were several factors that warranted my

recommendation for major revision in Results section of this manuscript.

On line 169 under the "FAT10 does not influence IL-12 expression in DC2.4 cells"

segment, there didn't appear to be any mention of validating successful lentiviral

transfection.

Reply: When the DC2.4 cells were successfully infected with lentivirus, they

overexpressed FAT10, and were named DC2.4-mFAT10. In Figure 1C one can see

that DC2.4-mFAT10 cells, in contrast to not-infected DC2.4 cells (Fig. 1A, C),

strongly express FAT10.This shows that DC2.4 cells were successfully infected

with lentiviruses.

In the next segment "FAT10 has no significant effect on IL-12 expression in bone

marrow-derived dendritic cells", BMDCs were stimulated with IFN-γ and LPS to induce

IL-12 expression. However, in the previous segment a combination IFNγ/TNF was used

to stimulate DC2.4s. The rationale behind using only IFN-γ as opposed to the

combination IFNγ/TNF wasn't immediately clear upon review.

Reply: In Figure 3, we only used IFNγ stimulation without TNF stimulation for the

following reason: To investigate whether FAT10 plays a role in IL-12 secretion in

BMDCs, we need FAT10 expression. Therefore, cells were stimulated with IFNγ for

24 h. After the induction of FAT10, IL-12 is induced with LPS. If we do the initial

FAT10 induction in the presence of TNF, TNF will mature BMDCs and therefore will

induce the expression of IL-12. However, to investigate a contribution of FAT10 in

IL-12 induction, we first need a FAT10 expression. As seen in Fig. 2A, FAT10expression in BMDCs is also strongly induced in the absence of TNF. We

explained this in the revised manuscript.

Lastly, in "FAT10 does not affect Th1 cell differentiation", the rationale as to why IFN-γ

was selected as the sole cytokine of focus was unclear. TNF is also considered an

important signature cytokine for Th1 differentiation, why was that not selected in addition

to IFN-γ to validate results?

Reply: IFN-γ is “the” signature cytokine to describe Th1 cells. For sure, TNF can

additionally be used to describe Th1 cells. However, expression of TNF correlates

with the expression of IFNγ. In our opinion, no substantial gain of information can

be obtained by including TNF in the analysis. Therefore, in our opinion it is not

justified to redo the experiments to include TNF.

Overall, while I do partly agree with the overall conclusions of this paper that FAT10 does

not appear to affect IL-12 signaling, I feel more validatory work is necessary to confirm

many of these results.

Reply: In our opinion, the experiments are very well controlled and solid.

Furthermore, we addressed the question whether FAT10 is involved in Il-12

signaling from many different aspects. Finally, all of the different setups provided

the same conclusive result. Namely, that FAT10 does not play a role in Il-12

signaling. Therefore, we strongly feel that our results are highly valid and

conclusive

---

## [Editor Report · Decision Letter 1]

1 Apr 2025

The ubiquitin-like modifier FAT10 does not affect IL-12 expression and signaling

PONE-D-25-01617R1

Dear Dr. Basler,

We’re pleased to inform you that your manuscript has been judged scientifically suitable for publication and will be formally accepted for publication once it meets all outstanding technical requirements.

Kind regards,

Mariola J Ferraro, Ph.D.

Academic Editor

PLOS ONE
---

## [Editor Report · Acceptance letter]

PONE-D-25-01617R1

PLOS ONE

Dear Dr. Basler,

I'm pleased to inform you that your manuscript has been deemed suitable for publication in PLOS ONE. Congratulations! Your manuscript is now being handed over to our production team.

Kind regards,

on behalf of

Dr Mariola J Ferraro

Academic Editor

PLOS ONE